# Optimizing Parametric Factors in CIELAB and CIEDE2000 Color-Difference Formulas for 3D-Printed Spherical Objects

**DOI:** 10.3390/ma15124055

**Published:** 2022-06-07

**Authors:** Ruili He, Kaida Xiao, Michael Pointer, Manuel Melgosa, Yoav Bressler

**Affiliations:** 1School of Design, Faculty of Arts, Humanities and Cultures, University of Leeds, Leeds LS2 9JT, UK; sdrh@leeds.ac.uk (R.H.); mrpointer@btinternet.com (M.P.); 2Optics Department, Faculty of Sciences, University of Granada, 18071 Granada, Spain; mmelgosa@ugr.es; 3Stratasys Ltd., Rehovot 76124, Israel; yoav.bressler@stratasys.com

**Keywords:** color difference, 3D object, visual assessment, CIELAB, CIEDE2000, optimization

## Abstract

The current color-difference formulas were developed based on 2D samples and there is no standard guidance for the color-difference evaluation of 3D objects. The aim of this study was to test and optimize the CIELAB and CIEDE2000 color-difference formulas by using 42 pairs of 3D-printed spherical samples in Experiment I and 40 sample pairs in Experiment II. Fifteen human observers with normal color vision were invited to attend the visual experiments under simulated D65 illumination and assess the color differences of the 82 pairs of 3D spherical samples using the gray-scale method. The performances of the CIELAB and CIEDE2000 formulas were quantified by the *STRESS* index and *F*-test with respect to the collected visual results and three different optimization methods were performed on the original color-difference formulas by using the data from the 42 sample pairs in Experiment I. It was found that the optimum parametric factors for CIELAB were $k_{L}$ = 1.4 and $k_{C}$ = 1.9, whereas for CIEDE2000, $k_{L}$ = 1.5. The visual data of the 40 sample pairs in Experiment II were used to test the performance of the optimized formulas and the *STRESS* values obtained for CIELAB/CIEDE2000 were 32.8/32.9 for the original formulas and 25.3/25.4 for the optimized formulas. The *F*-test results indicated that a significant improvement was achieved using the proposed optimization of the parametric factors applied to both color-difference formulas for 3D-printed spherical samples.

## 1. Introduction

With the rapid development of various 3D printing technologies, it has become more and more popular in recent years to produce colored solid objects using a 3D color printer because of its advantages of saving both time and money [1]. This method directly interconnects with advanced manufacturing techniques and customization with excellent accuracy and many applications have been involved, such as art and design practice [2], manufacture of soft tissue prostheses [3], dentistry [4], food [5], etc. Despite the advances in production technologies, faithfully reproducing the color appearance of 3D objects remains a challenge because it requires knowledge of 3D metrology, color modeling, human perception, and 3D printing [6]. The International Commission on Illumination (CIE) has listed the reproduction and measurement of 3D objects as one of the top priority topics in its current research strategy [7]. Furthermore, the CIE Technical Committee 8–17 has been established to develop methods for evaluating the color differences between 3D objects.

For measuring object colors, a CIELAB color space based on CIE XYZ tristimulus values was recommended by CIE [8] and the associated CIELAB color-difference formula has been widely used to quantify the perceived color difference between a pair of colored samples. Due to its visual non-uniformity, the CIEDE2000 color-difference formula was developed to improve the correlation between the computed and perceived color differences [9]. Currently, the CIEDE2000 color-difference formula is jointly recommended as a standard by the CIE and the International Organization for Standardization (ISO) [10].

The development of current color-difference formulas, however, has always been based on flat colored samples [11,12,13,14]. In contrast to a 2D sample, a 3D printed object has a non-uniform surface and it is more complicated for the human visual system to process color information, especially considering that the color appearance of 3D objects is probably affected by the 3D shape, gloss, and different lighting conditions [15,16,17]. It has been reported that color-difference formulas predict the lightness difference contribution to the total color difference in different ways for spherical and flat samples [17]. To understand the basic visual perception of colored 3D objects, Hung et al. conducted a series of psychophysical experiments to quantify the visual color differences of 3D objects using a 7-step gray scale [18] and it was found that chroma differences for high chroma or dark colors were not easy to visually assess. The authors suggested that in the color-difference formula, the measured lightness and chroma values must be further scaled to fit the visual data.

Considering that the $k_{L}$ parametric factor in CIEDE2000 is recommended to have a value of 1 under reference conditions, whereas for the textile industry $k_{L}$ = 2 was necessary to allow for the associated texture effect, the parametric factors in color-difference formulas have been used to improve the prediction performance in different applications. Liu et al. optimized the $k_{L}$ factor in the color-difference formulas CIELAB, CIEDE2000, CIE94, and CMC for digital images [19], and the results showed that CIEDE2000 (2.3:1) performed the best. Mirjalili et al. [13] concluded that optimizing the lightness parametric factor, $k_{L}$, resulted in an improvement in the performance of the CIEDE2000 formula.

In addition to modifying parametric factors, Huang et al. proposed that color-difference formulas modified by a power function provided better agreement with visually perceived color differences [11]. Jiang et al. found that using an additional power function with an exponent of 0.55 achieved an improvement (*STRESS* values in a range from 23.0 to 26.3) in the prediction of 150 pairs of 3D-printed samples compared to the results found using the original formulas (*STRESS* values in a range from 30.0 to 37.9) [17]. However, the authors suggested that the exponent of 0.55 probably over-fitted the experimental data. Moreover, Pan et al. optimized the exponents in the power functions applied to the color-difference formulas and reported that the addition of a power function gave better performance than optimizing the $k_{L}$ lightness factor for the 3D spherical samples [20].

The color difference evaluation of 3D objects has stimulated industrial and academic interest but currently, there is insufficient available color-difference data collected using 3D objects and comprehensive knowledge of the visual color perception of 3D objects is desired. In the present study, 42 pairs of spherical 3D-printed samples (Experiment I) were prepared to have predominantly lightness, chroma, or hue differences, and visual color-difference assessments were conducted to investigate the human color perception of these 3D objects in terms of the lightness, chroma, and hue dimensions. Moreover, considering the simplicity of CIELAB and the joint CIE/ISO recommendation of CIEDE2000 for industrial applications, these two color-difference formulas were tested using data derived from a further set of 40 pairs of spherical 3D-printed samples (Experiment II). 

## 2. Materials and Methods

### 2.1. 3D Sample Pairs

The 5 CIE-recommended color centers: gray, red, green, yellow, and blue [21] were used in this study to prepare 3D samples. Specifically, 5 colors with very similar chroma and hue but different lightness from the gray center and 10 colors with different lightness, chroma, or hue from each of the other four color centers were designed and the difference was within 10 units. By performing color characterization on the Stratasys J750 3D color printer, the 45 colors were printed with resins and 45 spherical samples with a diameter of 50 mm were prepared, as shown in Figure 1.

Each sample was measured at three different points using a Konica Minolta CM-700d spectrophotometer with d:8 geometry, small aperture (SAV 3 mm), and specular component-included (SCI) mode. The measured spectral reflectance data were used to calculate the corresponding CIELAB values using the CIE 1964 standard observer under the illumination used in this study. The average measurement results of each sample are shown in $L_{10}^{*}C_{ab,\;10}^{*}$ and $a_{10}^{*}b_{10}^{*}$ diagrams in Figure 2, where the triangle symbols are approximately positioned at the ‘centers of gravity’ of the 5–10 printed samples at each of the five CIE-recommended centers, gray, red, green, yellow and blue (Figure 1), and they represent the five samples that were considered as reference samples in Experiment II. To quantify the color homogeneity of the 3D-printed samples, the mean color difference from the mean (*MCDM*) of the CIELAB units was calculated from the results of the three measurements [17]. The measured average *MCDM* value of the 45 samples was 1.77 units, which demonstrates relatively good color uniformity using 3D printing technology [22].

One of the experimental designs, Experiment I, was intended to investigate the color difference between each sample of a 3D pair in terms of only one color attribute, that is, mainly lightness differences $\Delta L_{10}^{*}$, chroma differences $\Delta C_{ab,\;10}^{*}$, or hue differences $\Delta H_{ab,\;10}^{*}$. Thus, 9 testing pairs were produced for each of the four chromatic color centers to have predominant lightness/chroma/hue differences, such that the CIELAB lightness (or chroma or hue) difference for these pairs of samples was at least 85% of the total CIELAB color difference (see Figure 3). For the CIE gray center, only 6 testing pairs with predominantly lightness differences were produced (note that the gray samples had very small differences in hue and chroma). Overall, there were 42 pairs of 3D samples, including 18 lightness-difference pairs ($\Delta L_{10}^{*}/\Delta E_{ab,10}^{*}\;$ ≥ 0.85), 12 chroma-difference pairs ($\Delta C_{ab,10}^{*}/\Delta E_{ab,10}^{*}$ ≥ 0.85), and 12 hue-difference pairs ($\;\Delta H_{ab,10}^{*}/\Delta E_{ab,10}^{*}$ ≥ 0.85). In Experiment I, the color-difference magnitudes ranged from 2 to 9 CIELAB units and the average color difference was 5.46 CIELAB units.

The second experimental design, Experiment II, consisted of 40 pairs of spherical 3D samples with a common reference sample at each one of the five CIE color centers, shown by triangle symbols in Figure 2. Therefore, in Experiment II nine pairs for each one of the four CIE chromatic centers and four pairs for the CIE gray center were considered. The color differences of the 40 sample pairs in Experiment II ranged from 1 to 12 CIELAB units and the average color difference was 4.58 CIELAB units.

A total of 82 pairs of 3D samples were used in the visual experiments to test the performance of two color-difference formulas. Specifically, Experiment I aimed to investigate the visual perception of 3D samples in terms of lightness, chroma, and hue differences, the data to be used to optimize the color-difference formulas, and Experiment II was used to test the performance of the optimized formulas.

### 2.2. Gray-Scale Method

The gray-scale method has been widely used for visual assessment in previous studies due to its ease of use and accurate results [13,17,20,23,24]. In the present study, the *Grey scale for assessing change in colour* from the Society of Dyers and Colourists (SDC), following ISO 105-A02 [25], was used in the psychophysical experiments. This scale consists of 9 pairs of non-glossy neutral gray colored chips, with grades of 1, 1.5, 2, 2.5, 3, 3.5, 4, 4.5, and 5. The reflectance of each gray chip was measured using a CM700d spectrophotometer with SAV 3 mm and SCI, and the color difference between each pair of chips was calculated for the CIE1964 standard observer under the illumination used in the visual experiments. Figure 4 plots the CIELAB and CIEDE2000 color-difference values for the 9 grades on the gray scale. Grade 1 has the largest color difference (13.36 $\Delta E_{ab,10}^{*}$ or 13.13 $\Delta E_{00,10}$) and grade 5 has almost no color difference (0.18 $\Delta E_{ab,10}^{*}$ or 0.16 $\Delta E_{00,10}$). It can be observed in Figure 4 that the color-difference values calculated using the CIEDE2000 formula were slightly smaller than those of CIELAB, but the difference is not large for each grade because the color differences in the gray scale are predominantly caused by a lightness difference.

In order to quantify the relationship between the gray-scale grade values and the corresponding color differences, a third-order polynomial regression was used [24]. The fitted formulas for CIELAB and CIEDE2000 color-difference units are expressed in Equations (1) and (2), with *R^2^* values of 0.9998 and 0.9960, respectively. Since this study aimed to test and optimize the CIELAB and CIEDE2000 formulas, Equations (1) and (2) were used separately to transform the gray-scale grades (*GS*) reported by the human observers to visual color-difference values (Δ*V*).
(1)$$\Delta V_{ab,10}=-0.25*GS^{3}+3.05*GS^{2}-13.95*GS+24.46,$$
(2)$$\Delta V_{00,10}=-0.28*GS^{3}+3.36*GS^{2}-14.83*GS+24.81.$$

### 2.3. Visual Assessments

A VeriVide viewing cabinet with a D65 simulator was used in the visual color-difference assessments. The relative spectral power distribution of the D65 illumination was measured using a Konica Minolta CS2000 spectroradiometer and a reference white. The measured correlated color temperature, CIE color-rendering index $R_{a}$, and luminance at the center of the floor of the cabinet were 6519 K, 97, and 412.35 cd/m^2^, respectively.

The visual experiments were conducted in a dark room and human observers were asked to adapt to the dark surroundings for two minutes. Each pair of 3D samples was placed in the center of the viewing cabinet, as shown in Figure 5. The observer’s task was to evaluate the magnitude of the color difference in the pair in comparison with the perceived color differences in the pairs of the gray scale. All the testing pairs were presented in a random order and observers were encouraged to give intermediate assessment values with one decimal between two contiguous gray pairs (e.g., 3.6 for a color difference between the pairs 3.5 and 4 but closer to 3.5 than 4). During the visual experiments, observers had a fixed viewing position with an approximate distance of 50 cm from the samples and a 45° viewing angle. Before Experiment I commenced, a pilot experiment was performed to train the observers to make the visual assessments using the gray-scale method. In Experiments I and II, each observer repeated the visual assessments of all testing pairs three times.

A panel of 15 observers (10 females and 5 males) participated in the visual experiments, with ages ranging from 25 to 29. They were postgraduate students from the University of Leeds and had normal color vision according to the Ishihara test. Most observers had little experience in color-difference evaluation. A total of 3690 assessments (82 pairs × 3 repetitions × 15 observers) were conducted to collect visual color-difference data of 3D printed spherical sample pairs during the psychophysical experiments.

### 2.4. Metrics for Testing Color-Difference Formulas

#### 2.4.1. STRESS Index

The standardized residual sum of squares (*STRESS*) index (Equation 3) was proposed by García et al. [26] and adopted by CIE [27] to test the performance of two color-difference formulas with respect to a given set of visual color-difference data:(3)$$STRESS=100\sqrt{\left(\frac{\sum^{\;}{\left(\Delta E_{i}-f\Delta V_{i}\right)}^{2}}{\sum^{\;}\Delta E_{i}^{2}}\right)},$$
where $f=\frac{\sum^{\;}\Delta E_{i}\Delta V_{i}}{\sum^{\;}V_{i}^{2}}$, $\Delta E_{i}$ indicates the computed color difference of the ith testing pair by a color difference formula, e.g., for CIELAB, and $\Delta V_{i}$ is the corresponding average visual color difference for the same testing pair. The *STRESS* value ranges from 0 to 100, and for a perfect agreement, the *STRESS* value should be zero. The larger the *STRESS* value, the worse the agreement between perceived and computed color differences. 

The *STRESS* index was also used to compute intra- and inter-observer variability [28]. For each observer, intra-observer variability was computed as the average of *STRESS* values of each one of the 3 replications made by this observer with respect to the average result of the 3 replications, whereas inter-observer variability was computed as the *STRESS* value between the average result of the 3 replications of this observer and the average results of all 15 observers. Final intra- and inter-observer variability in the experiments were defined as the average intra- and inter-observer variability *STRESS* values from the 15 observers, respectively.

#### 2.4.2. F-Test

Although *STRESS* values can be used to compare the performance of two different formulas, it is not sufficient to indicate the degree of statistical significance. The *F-*test was used to analyze the statistical significances between the original and optimized color-difference formulas [26]: (4)$$F=\frac{STRESS_{A}^{2}}{STRESS_{B}^{2}}.$$

For the two-tailed *F*-distribution with a 95% confidence level, the critical value $F_{C}$ can be found from statistical tables with the degrees of freedom, $df_{A}$ and $df_{B}$, defined as $df_{A}=df_{B}=N-1$, where N is the number of sample pairs. The calculated F value from Equation (4) can be compared with the confidential interval [$F_{C},\;1/F_{C}$]. If the F value is smaller than $F_{C}$, it means that the optimized formula A is significantly better than the original formula B; if the F value is between $F_{C}$ and 1, the optimized formula A is insignificantly better than the original formula B; otherwise, there is no improvement achieved by formula A with respect to formula B.

### 2.5. Optimization Methods

In CIELAB, the total color difference is defined as a Euclidean distance in terms of lightness, chroma, and hue differences between the two stimuli. Most current advanced color-difference formulas were derived by modifying the CIELAB formula [9], following the generic Equation (5):(5)$$\Delta E=\sqrt{{\left(\frac{\Delta L_{10}^{*}}{k_{L}S_{L}}\right)}^{2}+{\left(\frac{\Delta C_{ab,10}^{*}}{k_{C}S_{C}}\right)}^{2}+{\left(\frac{\Delta H_{ab,10}^{*}}{k_{H}S_{H}}\right)}^{2}+\Delta R},\;$$
where $\Delta L_{10}^{*},\;\Delta C_{ab,10}^{*},\;\Delta H_{ab,10}^{*}$ are the CIELAB metric lightness, chroma, and hue differences, respectively, $k_{L},k_{C},\;k_{H}$ and $S_{L},S_{C},\;S_{H}$ are the three parametric factors and weighting functions for lightness, chroma, and hue differences, respectively, and ΔR is an interactive term between chroma and hue difference. In the CIELAB formula, $k_{L},k_{C},\;k_{H},\;S_{L},S_{C},\;S_{H}$ were all set as 1 and ΔR=0. However, in CIEDE2000, $S_{L},S_{C},\;S_{H}$ are three specified weighting functions, ΔR is related to the so-called rotation term affecting the blue saturated region of color space, and $k_{L}=k_{C}=k_{H}=1$ under so-called ‘reference conditions’ for most applications ($k_{L}=2$, $k_{C}=k_{H}=1$ for textiles). 

Considering that the color appearance of 3D objects may be affected by more factors than for 2D objects, it is hypothesized that values of the parametric factors in color-difference formulas should be different for these two situations. In order to improve the predictions of CIELAB and CIEDE2000 color-difference formulas for 3D printed spherical objects, three optimization methods were used in the current paper: 

Method 1: Optimize $k_{L}$ with $k_{C}=k_{H}=1$.Method 2: Optimize both $k_{L}$ and $k_{C}$ with $k_{H}=1$.Method 3: Apply a power function and optimize the exponent n (i.e., $\Delta E^{\prime }=\Delta E^{n}$).

The goal of the optimizations was to minimize the *STRESS* value between the visual results and the values calculated using a color-difference formula by using the GRG nonlinear method in Excel Solver or the *fminsearch* function in MATLAB. 

## 3. Results

### 3.1. Observer Variability

The observer variability was quantified by using the *STRESS* index [27] from visual color-difference values (ΔV) obtained from gray-scale grades reported by the observers participating in the experiments (see Equations (1) and (2)). The average *STRESS* values for the intra- and inter-observer variabilities were 30.7 and 30.9 CIELAB units (31.4 and 31.7 CIEDE2000 units), respectively. Similar results were achieved by Pan et al. where the intra- and inter-observer variabilities for matte spherical samples were 21.9 and 31.2 CIELAB units, respectively [20]. In comparison, the intra-observer variability in the present study was relatively larger, which is possibly because three repetitions were performed in this study by each observer. Moreover, Jiang et al. reported that the observer variability in the color-difference evaluation of 3D objects is slightly larger (23.5 CIELAB units for inter and 14.9 for intra) than that of flat 2D objects (19.4 CIELAB units for inter and 12.6 for intra) [17]. 

### 3.2. Visual Color Difference

The visual color-difference results (∆*V*) of the 82 pairs of 3D samples in Experiments I and II were plotted in Figure 6 against the corresponding color-difference values computed by the CIELAB and CIEDE2000 formulas. It shows that the visual data can be fitted as linear relationships (the dotted lines) to the computed color differences, with *R*^2^ values of 0.6244 and 0.6986 for CIELAB, Figure 6a, and CIEDE2000, Figure 6b, respectively. Moreover, the $\Delta E_{00,10}$ data have less scatter than the $\Delta E_{ab,10}^{*}$ data in Figure 6. It was to be expected that the scatter points should fall on the 45° dashed line if the color-difference formula can exactly predict the visual results and the larger the scatter, the worse the color-difference formula performs. In addition, the $\Delta E_{00,10}$ cluster points tend to be closer to the 45° dashed line than $\Delta E_{ab,10}$, especially within approximately 6 units of color-difference. This is in line with the statement that the CIEDE2000 formula was developed to fit visual assessment datasets of small-medium color differences, typically under five CIELAB color-difference units. 

Additionally, the ∆*V* magnitudes in Figure 6 are almost half (0.48) those of the calculated color-difference values. It was generally assumed that the ∆*V* values would tend to be close to the ∆*E* values if the color-difference formula is a good predictor of the visual data, but it has been reported that in most cases. the ratios instead of the absolute values of ∆*V* and ∆*E*, are helpful for testing color-difference formulas [29]. Poor correlation indicates that the color-difference formula should be improved to provide a better performance of color-difference assessment.

Considering the three color components in Equation (5), $\Delta L_{10}^{*},\;\Delta C_{ab,10}^{*},\;\Delta H_{ab,10}^{*}$, the 42 pairs of 3D samples in Experiment I were combined to keep the whole color difference predominantly from one attribute difference. Figure 7a shows the plots of ∆*V* against $\Delta L_{10}^{*},\;\Delta C_{ab,10}^{*},\;\Delta H_{ab,10}^{*}$ in Experiment I and three linear lines fitted for lightness, chroma, and hue differences. Figure 7b plots the visual data against the CIEDE2000-weighted $\Delta L_{00},\Delta C_{00},\;\Delta H_{00}$ calculated using the equations proposed by Nobbs [30].

As can be seen in Figure 7a, the relationships between ∆*V* and three components of $\Delta E_{ab,10}^{*}$ were approximately linear and the fitted hue-difference linear line is above the other two lines. This indicates that the perceived (∆*V*) hue differences are higher than the perceived (∆*V*) lightness differences for a given value of $\Delta E_{ab,10}^{*}$, which means that the human visual system is more sensitive to hue changes than to lightness changes in CIELAB. In comparison, the slope of the fitted chroma-difference line, as well as its $R^{2}$ value, are smaller than those of the lightness and hue-difference fitted lines, indicating that the sensitivity to perceived chroma changes (∆*V*) for 3D objects is relatively smaller. In addition, the CIELAB lightness, chroma, and hue differences of the 42 pairs are almost in the same range, from 2 to 10 units. The perceived visual color differences, however, have different ranges for these three components, which are approximately 1.0–3.0 units for chroma differences, 0.5–5.5 units for lightness differences, and 1.5–5.5 units for hue differences. Therefore, the factors related to these three color components in the color-difference formula should be rescaled for 3D color objects. 

In comparison, Figure 7b shows similar results for lightness and hue-difference predictions but quite different results for chroma differences where the CIEDE2000-weighted $\Delta C_{00}$ is much closer to the visual data. This verified that CIEDE2000 performed chroma correction on the CIELAB color-difference formula and an improvement was achieved.

### 3.3. Testing Color-Difference Formulas

The performances of CIELAB and CIEDE2000 color-difference formulas were tested using the visual data collected from the 42 pairs of samples in Experiment I, and the *STRESS* values obtained for CIELAB and CIEDE2000 were 28.6 and 25.9 units, respectively. These results show that CIEDE2000 has a slightly better performance than CIELAB for the color-difference prediction of 3D samples. 

In order to investigate the performance of CIELAB and CIEDE2000 formulas on predicting lightness, chroma, and hue differences, the *STRESS* values and the ratios of ∆*E*/∆*V* of the 42 pairs in Experiment I were calculated and the results are shown in Table 1 and Table 2, respectively. Ideally, the ratios ∆*E*/∆*V* should be 1.0 for perfect agreement between the predictions of a color-difference formula and experimental visual results.

Table 1 shows that the *STRESS* values in the CIEDE2000 units are smaller than those in the CIELAB units for the lightness-difference pairs and hue-difference pairs, but not for the chroma-difference pairs. Regarding the results in Table 2, both color-difference formulas have ratios different than 1.0 for predicting lightness difference, whereas the ratios for hue differences are the closest to 1.0, which is in agreement with the smallest *STRESS* values shown in Table 1. It is concluded that the tested color-difference formulas have better performance for hue differences than for lightness and chroma differences.

### 3.4. Optimization of Color-Difference Formulas

The 42 pairs of 3D samples in Experiment I were used to optimize the CIELAB and CIEDE2000 color-difference formulas using the three different methods described in Section 2.5. Table 3 gives the *STRESS* values and corresponding optimized factors for CIELAB and CIEDE2000. Table 4 lists the *F*-test results of the optimized formulas with respect to the original values. Concerning the degrees of freedom, which are 41 (N = 42), the critical value $F_{C}$ is 0.54 (1/$F_{C}=1.86$) for the two-tailed *F*-distribution with a 95% confidence level.

By using Method 1, which is to optimize the $k_{L}$ factor with $k_{C}=k_{H}=1$, the optimal $k_{L}$ factor for the CIELAB color-difference formula is 1.1 and the corresponding *STRESS* value is 28.5, which is similar to the *STRESS* value of 28.6 calculated using the original formula ($k_{L}=k_{C}=k_{H}=$1). Therefore, the improvement in the performance of the optimized CIELAB color-difference formula with $k_{L}=1.1$ is negligible. In comparison, the optimal $k_{L}$ factor for CIEDE2000 is 1.5 and the *STRESS* value reduced from 25.9 to 18.8 units, indicating much better performance than the original formula. Furthermore, the corresponding *F-*test value is 0.53 (Table 4), which is smaller than the critical value ($F_{C}$ = 0.54), showing that the optimized CIEDE2000 formula with $k_{L}$ = 1.5 is significantly better than the original formula. 

With Method 2 for optimizing both $k_{L}$ and $k_{C}$ simultaneously with $k_{H}$ = 1, the optimal $k_{L}$ and $k_{C}$ factors for CIELAB are 1.4 and 1.9, respectively, and the *STRESS* value between the optimized CIELAB and the visual data decreased from 28.6 to 20.5 units. Furthermore, the corresponding *F*-test value is 0.51, which is smaller than the $F_{C}$ value of 0.54, showing that the optimization of the CIELAB formula with $k_{L}$ = 1.4 and $k_{C}$ = 1.9 has significantly better performance than the original CIELAB formula. For the results of the optimized CIEDE2000 with $k_{L}$ = 1.6 and $k_{C}$ = 1.1, significantly better performance was also achieved compared to the original CIEDE2000 formula, with the *STRESS* value decreasing from 25.9 to 18.6 units and with an *F-*test value of 0.52. Moreover, the optimization results of CIEDE2000 using Method 1 ($k_{L}$ = 1.5, $k_{C}=k_{H}=1$) and Method 2 ($k_{L}$ = 1.6, $k_{C}$ = 1.1, $k_{H}=1$) are very similar because the $S_{C}$ function in CIEDE2000 has already corrected the CIELAB chroma difference values [9]. In contrast, the optimal $k_{C}$ factor in CIELAB is 1.9 to provide a better prediction performance.

With respect to the consequences of applying a power function to the original color-difference formulas (Method 3), the obtained *STRESS* values of the optimized CIELAB and CIEDE2000 formulas are 28.5 and 24.8, respectively, and the *F*-test values are very close to 1. When the power function was applied to the optimized color-difference formulas, named Method 1 + 3 and Method 2 + 3 in Table 3 and Table 4, results similar to those from Method 1 and Method 2 were obtained. This means that applying a power function gives almost no improvement on the optimization of color-difference formulas for 3D samples. 

Additionally, each one of the three kinds of color pairs in Experiment I (i.e., pairs with ≥85% lightness, chroma, and hue-differences) were used to fit the $k_{L},\;k_{C},\;k_{H}$ parametric factors separately in CIELAB and CIEDE2000. The results and corresponding *STRESS* values are listed in Table 5. The optimal $k_{L}$ factors (1.8 and 1.6) for both formulas are larger than the default value of 1.0, indicating that lightness differences in the original formulas were over-valued (see Equation (5)) for 3D samples. The optimal results for the $k_{C}$ and $k_{H}$ factors are quite different for CIELAB and CIEDE2000, for example, the optimal $k_{C}$ value is 1.4 for CIELAB but 0.5 for CIEDE2000, as opposed to the optimized results for the $k_{H}$ factor. The *STRESS* values in Table 5 show that the optimization of CIEDE2000 is better than that of CIELAB, particularly for the lightness differences. Compared with the results calculated using the original formulas (Table 1), the *STRESS* values after individual factor optimization become smaller, which implies that the approach chosen, i.e., to optimize the parametric factors using separate datasets, did not yield significantly improved results.


### 3.5. Testing the Optimized Color-Difference Formulas

In addition to reporting the results of the optimized color-difference formulas based on the 42 pairs of 3D samples in Experiment I that were used as the training data for optimization, the collected visual color-difference data of the 40 pairs of 3D samples in Experiment II were also used to test the performance of the optimized formulas and the *STRESS* values obtained are shown in Table 6. Table 7 gives the *F*-test results of the optimized formulas with respect to the original ones. Given the degrees of freedom, 39 (N = 40), the critical value $F_{C}$ is 0.53 (1/$F_{C}=1.89$) for the two-tailed F-distribution with a 95% confidence level. 

It is noticeable in Table 6 that by using Method 2 to optimize the CIELAB formula, the *STRESS* value decreased from 32.8 to 25.3 units, whereas for the optimizations on CIEDE2000, Methods 1 and 2 gave similar small *STRESS* values that decreased from 32.9 to 26.0 and 25.4, respectively. The effect produced by the optimized $k_{C}$ factor in CIELAB is not necessary for CIEDE2000 because it is already produced by the weighting function for the chroma ($S_{C}$) of CIEDE2000. However, $k_{L}$ optimized factors are useful both in CIELAB and in CIEDE2000. Furthermore, the corresponding *F*-test values in Table 7 show that the performance of the optimized color-difference formulas was greatly improved.

## 4. Discussion

The purpose of this study was to investigate human color perception of the lightness, chroma, and hue differences of 3D spherical objects and to optimize the current CIELAB and CIEDE2000 color-difference formulas using parametric factors and visual results collected from psychophysical experiments. It was found that it is generally easier to assess the hue differences of 3D spherical objects but not the chroma differences, and the results indicated that the parametric factors related to the lightness differences, chroma differences, and hue differences in color-difference formulas should be optimized for 3D objects. 

Among the three optimization methods tested, the best performance achieved for CIELAB was to optimize both the $k_{L}$ and $k_{C}$ parametric factors, and the optimal $k_{C}$ factor (1.9) is larger than the optimal $k_{L}$ factor (1.4), indicating that the original CIELAB formula predicted a larger difference for the chroma dimension than for the lightness dimension. Moreover, both optimized factors are larger than the original values of 1.0, suggesting that the difference scale should be compressed for 3D objects in the CIELAB color-difference formula. In addition, the method of optimizing only the $k_{L}$ factor gives little improvement to CIELAB, which is quite contrary to the CIEDE2000 formula. 

The optimal $k_{L}$ parametric factor for the optimized CIEDE2000 is 1.5, which is larger than the default value of 1.0, suggesting that the visual lightness difference of 3D objects is over-estimated by the original formula. Perhaps this is because the CIEDE2000 formula with $k_{L}=1$ was developed based on 2D samples with homogeneous surfaces, whereas 3D objects have non-uniform surfaces and human color perception can be easily affected by other factors such as 3D shape, gloss, and shadows, etc. Similarly, in the textile industry, it is common practice to set the lightness parametric factor to 2 [14]. However, the experimental conditions leading to this parametric correction to lightness-difference sensitivity are not yet well understood; Liu et al. proposed $k_{L}$ = 2.3 for assessing color differences in digital images [19], and Huertas et al. investigated the three parametric factors based on simulated random-dot textures, suggesting values which were always larger than 1.0 [31]. 

In comparison to the optimization of the parametric factors, the power function had no evident improvement over the original formulas in this study. A possible reason for this is that the 3D sample pairs used in the current visual experiments had small to medium color differences ranging from 2 to 9 CIELAB units. In the study of Jiang et al. [17], a remarkable improvement was achieved by adding a power correction in predictions of the color differences between 3D objects in a range of 25 CIELAB units, and it was reported that the color-difference magnitude had more effect on the perceived color differences of 3D objects than the sample shape or illumination. Therefore, a power function is possibly more suitable for sample pairs with magnitudes in a very large range of color differences. 

## 5. Conclusions

This study conducted psychophysical experiments using 3D-printed spherical samples with the gray-scale method to assess the color differences. The visual color-difference results indicate that human color perception of 3D objects is different in lightness, chroma, and hue differences, and the factors related to these three components in the color-difference formulas need to be optimized for 3D objects. By using different optimization methods in CIELAB and CIEDE2000 formulas, it was found that power functions do not improve predictions of visual results in the current Experiments I and II as has been the case with previous experiments reported in the literature, perhaps because the magnitude of color differences was larger than in the current experiments. For the CIELAB color-difference formula, both the $k_{L}$ and $k_{C}$ factors should be optimized simultaneously, and considerable improvement was achieved with $k_{L}$ = 1.4 and $k_{C}$ = 1.9; specifically, the *STRESS* value of 40 testing sample pairs decreased from 32.8 to 25.3 units. For the optimization of the CIEDE2000 color-difference formula, $k_{L}$ = 1.5 is recommended for 3D spherical objects, which means that the *STRESS* value for the mentioned 40 testing pairs decreased by 7 units compared with the results from the original formula ($k_{L}$ = 1). 

## Figures and Tables

**Figure 1 materials-15-04055-f001:**
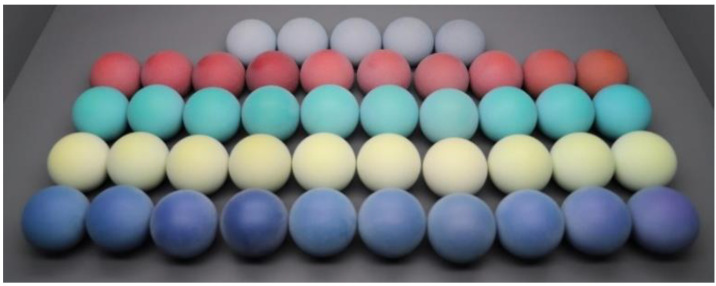
The 45 spherical samples printed using the Stratasys J750 3D color printer.

**Figure 2 materials-15-04055-f002:**
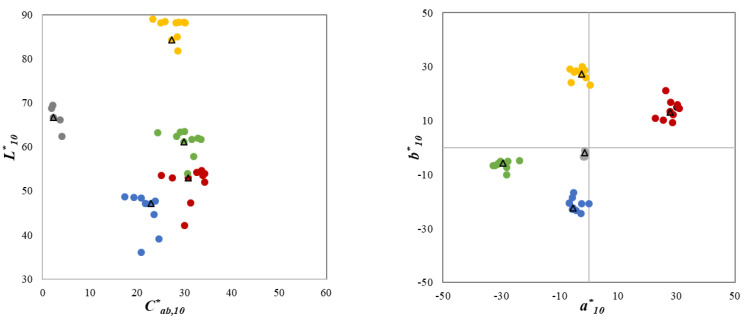
Color distributions of the 45 spherical samples in $L_{10}^{*}C_{ab,\;10}^{*}$ and $a_{10}^{*}b_{10}^{*}$ plane.

**Figure 3 materials-15-04055-f003:**
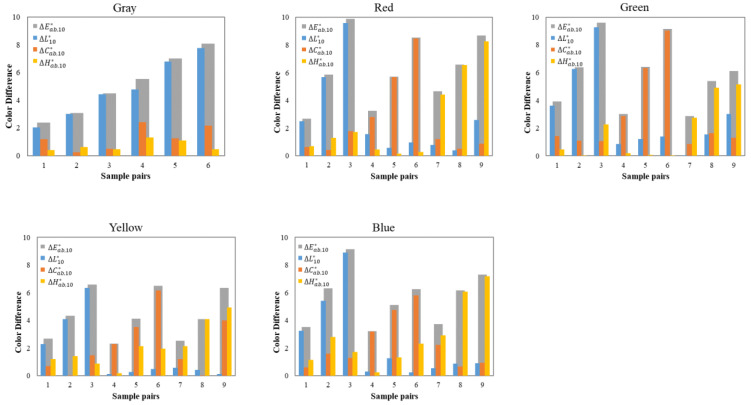
Values of $\Delta L_{10}^{*},\;\Delta C_{ab,10}^{*},\;\Delta H_{ab,10}^{*},\;\Delta E_{ab,10}^{*}$ for the 42 pairs of 3D samples in Experiment I.

**Figure 4 materials-15-04055-f004:**
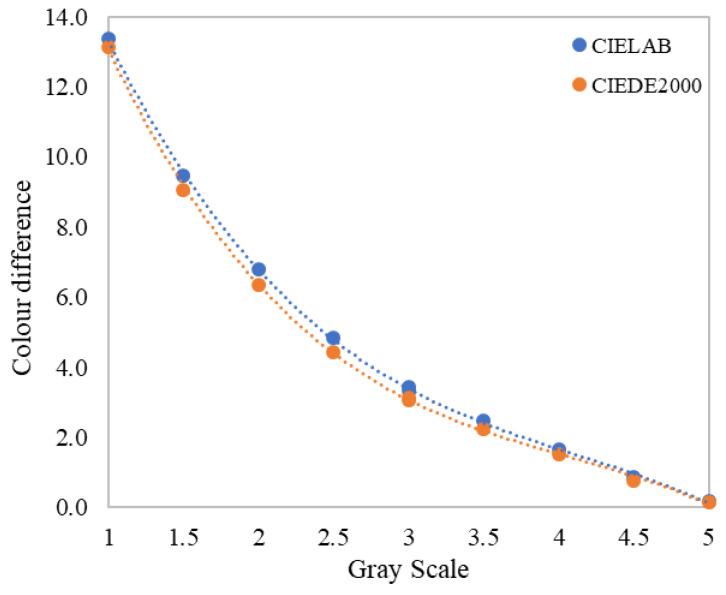
Relationship between the grade values in the 9 color pairs of the SDC gray scale for change in color and their measured color differences in CIELAB and CIEDE2000 units.

**Figure 5 materials-15-04055-f005:**
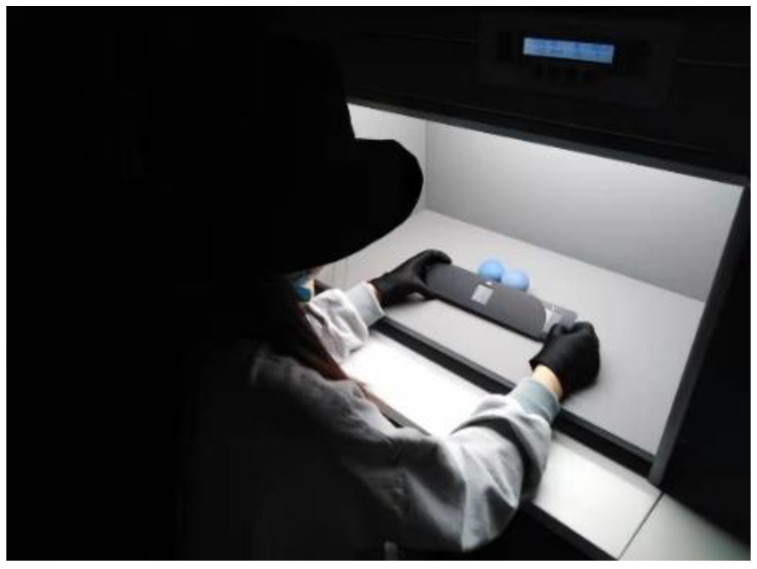
The visual color-difference assessment.

**Figure 6 materials-15-04055-f006:**
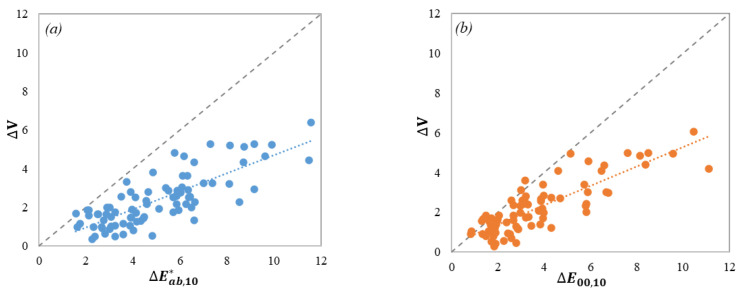
Plots of Δ*V* against (**a**) $\Delta E_{ab,10}^{*}$ and (**b**) $\Delta E_{00,10}$ for the 82 sample pairs.

**Figure 7 materials-15-04055-f007:**
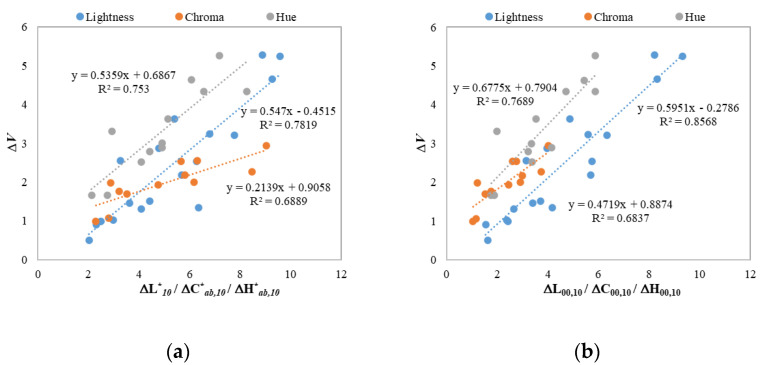
Plots and correlations of Δ*V* against $\Delta L_{10}^{*},\;\Delta C_{ab,10}^{*},\;\Delta H_{ab,10}^{*}$ (**a**) and $\Delta L_{00},\Delta C_{00},\;\Delta H_{00}$ (**b**) of the 42 sample pairs in Experiment I.

**Table 1 materials-15-04055-t001:** *STRESS* values in CIELAB and CIEDE2000 units of the 42 pairs with predominant (≥85%) lightness, chroma, and hue-differences (Experiment I).

	$\Delta L_{10}^{*}$ Pairs	$\Delta C_{ab,\;10}^{*}\;\mathrm{Pairs}$	$\Delta H_{ab,\;10}^{*}\;\mathrm{Pairs}$
CIELAB	23.7	21.7	17.7
CIEDE2000	18.9	23.3	15.6

**Table 2 materials-15-04055-t002:** Basic statistical results of ΔE/Δ*V* ratios of the 42 pairs with predominant (≥85%) lightness, chroma, and hue differences (Experiment I).

	$\Delta L_{10}^{*}$ Pairs		$\Delta C_{ab,\;10}^{*}\;\mathrm{Pairs}$		$\Delta H_{ab,\;10}^{*}\;\mathrm{Pairs}$	
Ratio	$\Delta E_{ab,10}^{*}$/Δ*V*	$\Delta E_{00,10}$/Δ*V*	$\Delta E_{ab,10}^{*}$/Δ*V*	$\Delta E_{00,10}$/Δ*V*	$\Delta E_{ab,10}^{*}$/Δ*V*	$\Delta E_{00,10}$/Δ*V*
Min	1.38	1.38	1.51	0.79	1.12	0.96
Max	4.91	4.37	3.76	1.99	2.19	1.59
Mean	2.66	2.41	2.63	1.44	1.63	1.26
STD	0.92	0.71	0.60	0.39	0.28	0.19

**Table 3 materials-15-04055-t003:** The *STRESS* values and corresponding optimized parametric factors for CIELAB and CIEDE2000 formulas, considering the visual results from Experiment I.

	Original $(k_{L}$$=k_{C}$$=k_{H}$ = 1)	Method 1	Method 2	Method 3	Method 1 + 3	Method 2 + 3
CIELAB	28.6	28.5 $(k_{L}$ = 1.1)	20.5 $(k_{L}$ $=1.4,\;k_{C}$ = 1.9)	28.5 (*n* = 0.9)	28.4 $(k_{L}$ = 1.1, *n* = 1.0)	20.5 $(k_{L}$ $=1.4,\;k_{C}$ = 1.9, *n* = 1.0)
CIEDE2000	25.9	18.8 $(k_{L}$ = 1.5)	18.6 $(k_{L}$ $=1.6,\;k_{C}$ = 1.1)	24.8 (*n* = 0.8)	18.7 $(k_{L}$ = 1.6, *n* = 1.0)	18.4 $(k_{L}$ $=1.6,\;k_{C}$ = 1.2, *n* = 0.9)

**Table 4 materials-15-04055-t004:** *F*-test values of the optimized CIELAB and CIEDE2000 formulas with respect to the original ones for visual results from Experiment I. Bold numbers indicate cases with statistically significant improvements.

	Method 1	Method 2	Method 3	Method 1 + 3	Method 2 + 3
Optimized CIELAB	0.99	**0.51**	1.00	0.99	**0.51**
Optimized CIEDE2000	**0.53**	**0.52**	0.92	**0.52**	**0.51**

**Table 5 materials-15-04055-t005:** The individual optimized $k_{L},\;k_{C},\;k_{H}$ factors and corresponding *STRESS* values (in parentheses) for CIELAB and CIEDE2000, considering three kinds of color pairs in Experiment I.

	$$\Delta L_{10}^{*}\;\mathrm{Pairs}$$	$$\Delta C_{ab,\;10}^{*}\;\mathrm{Pairs}$$	$$\Delta H_{ab,\;10}^{*}\;\mathrm{Pairs}$$
CIELAB	$k_{L}$ = 1.8 (23.3)	$k_{C}$ = 1.4 (21.6)	$k_{H}$ = 0.5 (16.7)
CIEDE2000	$k_{L}$ = 1.6 (18.3)	$k_{C}$ = 0.5 (21.3)	$k_{H}$ = 1.1 (15.5)

**Table 6 materials-15-04055-t006:** The *STRESS* values of the original and optimized color-difference formulas using the 40 sample pairs in Experiment II.

	Original	Method 1	Method 2	Method 3	Method 1 + 3	Method 2 + 3
CIELAB	32.8	32.6	25.3	33.1	33.0	25.3
CIEDE2000	32.9	26.0	25.4	31.2	26.0	25.1

**Table 7 materials-15-04055-t007:** The *F*-test values of the optimized formulas using the 40 sample pairs in Experiment II.

	Method 1	Method 2	Method 3	Methods 1 + 3	Methods 2 + 3
CIELAB	0.99	0.59	1.02	1.01	0.59
CIEDE2000	0.62	0.60	0.90	0.62	0.58

## Data Availability

The data presented in this study are openly available in Zenodo at https://doi.org/10.5281/zenodo.6502724 (accessed on 7 May 2022).

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
