# Peer review of "Optimizing Parametric Factors in CIELAB and CIEDE2000 Color-Difference Formulas for 3D-Printed Spherical Objects"

_materials, 2022, doi:10.3390/ma15124055_

Round 1
Reviewer 1 Report
The paper entitled "Optimizing parametric factors in CIELAB and CIEDE2000
color-difference formulas for 3D printed spherical objects" is written well and organized properly. The work presented in the paper is novel and benefits researchers in the area of 3D printing. The following minor comments need to be addressed before publication.
1) Why authors used the grayscale method to assess colour differences?
2) In line 439, the statement "textile samples considering the complex texture of textile samples" Kindly elaborate on the complex textile texture along with a reference.
Reviewer 2 Report
In this paper, authors test and propose an optimization of the CIELAB and CIEDE2000 color-difference formulas when applied to 3D printed samples.
The paper is very well written and structured, it undoubtedly catches the reader's attention and explains very well how the experiment was conducted, its results and how these could be interpreted.
I just have some minor suggestions that the author's may want to consider:
- In Conclusions, I have missed some references to future developments of this line of research, ¿what new questions arise from the results of this experiment?¿what would be the technical fields for its inmediate application?
- Also, I would recommend the authors to highlight paper weaknesses ¿maybe the number of participants could be a bit low (just 15), maybe the "unbalanced" number of women/men may have an influence in perception results...?
Reviewer 3 Report
1. The manuscript gives clear answers about the optimal parameters of commonly used color difference formulae for evaluating visual differences of 3D objects. The finding not only meets the academic requirements but also has practical application value.
2. The manuscript is easy to understand, and the quality of literature review, experimental design and data analysis is solid.
3. The introduction about how to perturb the lightness, chroma and hue of color samples is not clear enough to readers. Does the software of the printer allow CIELCh-based color assignment?
4. In my understanding, the output of Equ.1 and Equ.2 are approximation of dE*ab and dE00 measurement respectively. The curves are shown in Figure 4. I don’t think it should be called dV which represents visual color-difference values. dV should be obtained by visual experiments, not the color measurement using CM-700d spectrophotometer.
5. I suggest adding a figure which is similar to Figure 7, but the horizontal scales are CIEDE2000 scaled dL, dC, dH.
6. There is a need to discuss the dV differences between the gray samples and the high chroma samples.
7. In Table 3, I suggest adding (??=KC=1.0) for the “Original” sets.
8. In your visual experiment, the gray scales were observed against black background but the color samples used gray background instead. As the main idea of lightness correction in CIEDE2000 is taking the Crispening Effect of gray background into account, do you think Crispening Effect would affect the visual judgement to some degree?
